# DNA Damage Clustering after Ionizing Radiation and Consequences in the Processing of Chromatin Breaks

**DOI:** 10.3390/molecules27051540

**Published:** 2022-02-24

**Authors:** Veronika Mladenova, Emil Mladenov, Martin Stuschke, George Iliakis

**Affiliations:** 1Clinic and Polyclinic for Radiation Therapy, Medical School, University of Duisburg-Essen, 45122 Essen, Germany; veronika.mladenova@uk-essen.de (V.M.); emil.mladenov@uk-essen.de (E.M.); martin.stuschke@uk-essen.de (M.S.); 2Institute of Medical Radiation Biology, Medical School, University of Duisburg-Essen, 45122 Essen, Germany

**Keywords:** ionizing radiation, high-LET ionizing radiation, charged-particle radiotherapy, double-strand breaks, DNA damage response, complex double-strand breaks, double-strand break clusters, protons, heavy ions, ATM, ATR

## Abstract

Charged-particle radiotherapy (CPRT) utilizing low and high linear energy transfer (low-/high-LET) ionizing radiation (IR) is a promising cancer treatment modality having unique physical energy deposition properties. CPRT enables focused delivery of a desired dose to the tumor, thus achieving a better tumor control and reduced normal tissue toxicity. It increases the overall radiation tolerance and the chances of survival for the patient. Further improvements in CPRT are expected from a better understanding of the mechanisms governing the biological effects of IR and their dependence on LET. There is increasing evidence that high-LET IR induces more complex and even clustered DNA double-strand breaks (DSBs) that are extremely consequential to cellular homeostasis, and which represent a considerable threat to genomic integrity. However, from the perspective of cancer management, the same DSB characteristics underpin the expected therapeutic benefit and are central to the rationale guiding current efforts for increased implementation of heavy ions (HI) in radiotherapy. Here, we review the specific cellular DNA damage responses (DDR) elicited by high-LET IR and compare them to those of low-LET IR. We emphasize differences in the forms of DSBs induced and their impact on DDR. Moreover, we analyze how the distinct initial forms of DSBs modulate the interplay between DSB repair pathways through the activation of DNA end resection. We postulate that at complex DSBs and DSB clusters, increased DNA end resection orchestrates an increased engagement of resection-dependent repair pathways. Furthermore, we summarize evidence that after exposure to high-LET IR, error-prone processes outcompete high fidelity homologous recombination (HR) through mechanisms that remain to be elucidated. Finally, we review the high-LET dependence of specific DDR-related post-translational modifications and the induction of apoptosis in cancer cells. We believe that in-depth characterization of the biological effects that are specific to high-LET IR will help to establish predictive and prognostic signatures for use in future individualized therapeutic strategies, and will enhance the prospects for the development of effective countermeasures for improved radiation protection during space travel.

## 1. Introduction

Cancer is a leading cause of death worldwide, accounting for nearly ten million deaths in 2020. This high death toll has necessitated the development of effective therapies. Indeed, a variety of therapies have been developed and successfully applied in the clinic, resulting in improved patient quality of life and cures for some malignancies [1]. Notably, radiotherapy remains an integral part of most cancer treatment regimens. It utilizes ionizing radiation (IR) that exerts strong cytostatic and cytotoxic effects that are more prominent in rapidly proliferating malignant cells than in normal cells [2].

The modern applications of ionizing radiation date back to 1895 with the pioneering discovery of X-rays by Wilhelm Roentgen that were complemented by the discovery of radioactivity by Henri Becquerel and his doctoral student—Marie Skłodowska Curie. Their work with radioactive materials, although initiated as “one of pure science, done for itself, for the beauty of science” not only secured for Becquerel and Curie Nobel Prizes in Physics, but it also helped to reveal the diagnostic and therapeutic potential of IR, thus building the foundations of contemporary radiotherapy.

Reports dating as far back as 1902–1904 corroborate the application of radium in treating neck carcinomas and delivering radiation through glass tubes placed in close proximity to the tumors—a precursor of interstitial brachytherapy [3]. At present, photon beam radiotherapy in the form of low-LET X-rays is the standard and most widely used treatment in clinical settings worldwide [4].

Over the past decade, a number of considerable technological advances, such as image-guided radiotherapy, intensity-modulated photon radiotherapy and stereotactic body radiotherapy, have enabled more precise delivery of the required dose to the tumor, while minimizing the exposure of the surrounding normal tissues [5]. However, due to its physical properties, thoroughly discussed in the second section, low-LET IR deposits part of its energy outside the target volume. Additional advances are required to further minimize the inherent normal tissue toxicity and to improve tumor control and overall survival.

In an effort to reduce the side-effects of photon beams, proton beams have been tested in proton beam radiotherapy, which subsequently led to the ever-increasing interest in CPRT utilizing higher LET modalities, including carbon ion radiotherapy (CIRT). At present, there is a lot of excitement in the use of high-LET IR as modality to treat human cancer [6]. It is fostered by the need to address health risk issues associated with space exploration, which derives from the presence of highly energetic charged particles in the space environment. However, despite the obvious benefits and the increased spectrum of opportunities provided by high-LET IR modalities, there are potential risks and concerns associated with their use, which need to be carefully addressed.

In proton beam radiotherapy, the year 1946 marks its initial conception by Robert R. Wilson, followed by the treatment of the first patient in 1954 at the Lawrence Berkeley National Laboratory [7]. In 1961, a proton facility begun to operate at Harvard that remains operational to this date [8]. CPRT is considered beneficial for cancer patients because it deposits the IR-dose more specifically to the tumor, while minimizing the dose delivered to normal tissue. At present, the application of advanced techniques, such as 360° rotational gantries and intensity-modulated proton therapy, offers further improvements and aims to further minimize normal tissue toxicity [9].

Over the past decade, there has also been increasing interest in the application of heavy ions (HI) beams in radiotherapy. The first HI therapy program was initiated in 1975 by the Lawrence Berkeley National Laboratory group utilizing the BEVELAC accelerator [10]. At the same location radiotherapies with helium ions, carbon ions, neon ions, argon ions and silicon ions were conducted for the first time in 1975, 1977, 1977, 1979 and 1982, respectively [11,12]. Among the ions tested, carbon ions showed the most favorable biological characteristics and CIRT has become standard in the treatment of cancer patients using HI.

CIRT has been significantly developed and enriched during the last several years with layer-stacking methods, ultra-fast pencil-beam scanning techniques, etc. [13,14]. According to information summarized on the website of the particle therapy co-operative group (PTCOG), from 1954 to 2020, 248,384 and 39,210 patients underwent proton beam radiotherapy and CIRT, respectively [15]. Technological developments along with information on cancer’s pathophysiology have made possible greater conformity, optimized treatment planning and substantial increases in the dose delivered to the tumor. PTCOG reports that currently, more than one hundred facilities utilizing protons or carbon ions are in clinical operation worldwide, nearly forty are under construction and twenty-nine are at planning stages.

## 2. The Physical Characteristics of Low-Versus High-LET IR Determine Differences in Complexity of Induced DSBs

It is well-established that low-LET IR modalities, such as γ-rays or X-rays, generate various forms of DNA lesions through either direct ejection of an electron from the DNA, or attack by a hydroxyl radical (^•^OH) produced by the radiolysis of water [16,17].

The wide spectrum of generated DNA lesions includes sugar and base damages, as well as DSBs that are induced at ratios of about 20:1 [18]. Some forms of sugar damage disrupt the phosphodiester backbone of the DNA molecule and produce single-strand breaks (SSBs). If such SSBs are located in close proximity on the opposite DNA strands with a maximum separation of less than 10 bp, DSBs are generated. However, this mechanism of DSB induction is rare in mammalian cells, as it would have resulted in dose-response curves that show a linear quadratic increase in DSB yield with increasing IR dose. Actually, in most cases, the yield of DSBs in cells increases linearly with dose [19]. This is because DSBs are mainly produced by ionization clusters generated at the ends of tracks of secondary electrons produced by X-rays or γ-rays. Thus, DSBs are induced by events that increase in numbers linearly with IR-dose resulting in a linear dose-response curve [20].

When ionization clusters hit the DNA, they can evoke damage on both strands of the double helix and, thus, give rise to DSBs. The prevailing assumption in the field is that the adverse biological effects of X-rays or γ-rays derive from DSBs generated within such ionization clusters [21,22], rather than by independently generated ionizations on opposite DNA strands. Despite the generation of ionization clusters at the ends of low energy electron tracks, X-rays and γ-rays still deposit 50–70% of their energy in distinct ionization events. The latter derive from high-energy electrons that ionize sparsely and generate a relatively even ionization pattern within the cell [21,22]. This is why X-rays and γ-rays are considered sparsely ionizing forms of IR (Figure 1).

When IR beams are used to treat cancers that are located deep inside the body, a characteristic of the radiation modality that becomes important is the pattern of energy absorption with increasing depth of absorbing material. In the case of X-rays or γ-rays, there is a small build-up in absorbed dose at low depths followed by an exponential decline with increasing depth (Figure 2). This form of energy deposition is obviously suboptimal for treating deep-seated tumors [23,24].

In contrast to photons, forms of IR comprising charged particles, such as α-particles, or carbon ions ionize densely along their tracks (Figure 1). Moreover, the energy transferred to the absorbing material increases as the particle slows down and is at maximum just before the particle stops. This pattern of energy deposition is described by the “Bragg peak” [24]. Examples of Bragg peaks for different charged particles are shown in Figure 2. The increase in dose deposition afforded by charged particles at depths of absorbing material, which depend on particle energy, offers distinct advantages in the treatment of deep-seated tumors.

In practice, however, Bragg peaks are too narrow to treat tumors with dimensions of several centimeters—as is the case for the majority of human tumors. Therefore, methods have been developed to modulate particle energy at the entrance window and to generate a “spread-out Bragg peak” (SOBP), as the sum of multiple individual Bragg peaks generated by beams of varying energy. In the SOBP, a uniform dose can be administered to the target tumor volume, while still sufficiently sparing the surrounding normal tissue. In addition to the beneficial energy deposition characteristics of charged particles, particularly HI, an additional advantage is that they can be magnetically focused and that they show reduced lateral scattering. In this way, different field forms and sizes can be generated for an optimal tumor treatment.

There are also distinct biological advantages associated with the use of HI. Enhanced ionization clustering generated by high-LET particles, as compared to secondary electrons produced by low-LET X-rays, correlates with increased complexity/clustering of the damage induced in the DNA (Figure 1). Complex damage comprises at least two DNA lesions within one helical turn of DNA. This form of damage is referred to in the literature as clustered damage sites (CDS) or multiply damaged sites (MDS) [21]. From this form of lesion, DSBs can be generated. The proportion of CDS to single lesions induced in a cell increases with increasing LET of the radiation modality employed [16,25]. Some forms of CDS consist of base damage and may not immediately transform to DSBs (non-DSB clusters). For instance, a non-DSB cluster, designated as a bistranded cluster, comprises two apurinic/apyrimidinic (AP) sites on the opposite strands, or an SSB opposing an AP site. It could be converted to a delayed DSB through an incision of the AP site during base excision repair (BER) before removal of the SSB. Other CDS may contain SSBs and may transform to DSBs during processing of the base lesion after irradiation or during DNA replication [26,27,28,29]. In addition, IR has the potential to induce thermally labile sugar lesions that could be converted to DSBs as a result of temperature-dependent chemical processing during the incubation of cells at 37 °C [19]. Such chemical transformation of lesions to SSBs within a CDS will generate additional DSBs shortly after IR [21,30,31,32]. Notably, we have shown that the transformation of thermally labile sugar lesions to SSBs varies among cell lines and is predictive of cellular radiosensitivity [33].

Arguably, the most severe form of complex DNA damage is the clustered DSB (DSB cluster), which represents individual DSBs induced in close proximity (Figure 1). Each DSB within a cluster can be generated in any of the aforementioned ways. DSB clusters will be highly destabilizing for chromatin, where all DSB processing takes place and may cause the detachment from the genome of entire segments that can reach the size of chromatin loops, depending on the distribution of DSBs within the cluster (Figure 1). DSB clusters can be considered as a form of chromothripsis: A single, local catastrophic event characterized by extensive chromatin fragmentation that causes gross genomic rearrangements [34,35]. This phenomenon jeopardizes the efficient DSB processing and evokes inaccurate rejoining that fuels carcinogenesis [16,36]. Several studies dating from the mid–late 1990s suggest that DSB clusters are repaired inefficiently [37,38,39]. DSB clusters have often been considered as particularly consequential in mathematical models of IR action. Such models and Monte Carlo simulations suggest that they cause short DNA fragment-loss (peak at 85 bp) with probability that increases with decreasing fragment length [40,41], which enhances cell lethality [42,43,44,45].

In our previous studies, we modeled such DSB clustering and investigated its consequences on chromatin, by introducing enzymatic (I-SceI) induction of individual DSBs and DSB clusters of increasing complexity using a defined model system [36]. The results presented in this report confirm that increased DSB clustering correlates with increased cell lethality. Notably, enhanced DSB cluster formation is also considered a key characteristic of high-LET IR (see above). The prevailing hypothesis is that they are a major determinant of the higher relative biological effectiveness (RBE) of high-LET ionizing radiation modalities. RBE is defined as the ratio of doses between reference radiation (250 kV X-rays) and test radiation, inducing the same biological effect.

CIRT, compared to conventional photon beam radiotherapy, is also shown to induce irreparable DSBs manifesting in enhanced cell lethality in a glioma patient-derived stem and non-stem cells [46] and in neuroblastoma and glioblastoma cell lines [47]. Accumulated data from atomic force microscopy imaging confirm the induction of clustered DSBs and the formation of short DNA fragments—even when irradiating “naked” DNA lacking any higher order of chromatin organization [48]. The latter set of experiments also confirms that the size of the generated DNA fragments is inversely proportional to the LET of radiation used.

The production of short DNA fragments with increasing LET has also been associated with increased cellular toxicity and lethality. The latter is attributed to the impaired bilateral binding of KU70/80-heterodimer, a key component of classical non-homologous end-joining (c-NHEJ), on such very short fragments [49]. In line with these findings, it has also been shown that short DNA fragments (14–20 bp) impede the activity of the DNA-PK holoenzyme [48]. In aggregate, available results suggest that high-LET IR compromises the efficiency of c-NHEJ, which causes a switch towards resection-dependent DSB repair pathways. These aspects of high-LET IR will be further discussed below.

The distinct physical properties of low- and high-LET IR have also been demonstrated at the cellular level by applying immunofluorescence analysis that detects the recruitment of γ-H2AX and 53BP1 to sites of DSBs. A recent report studies DSBs generated by boron and neon ions accelerated to similar LET values (~135 keV/μm) at low energies (8 and 47 MeV per nucleon, respectively) [50]. High-resolution confocal microscopy reveals in this study specific patterns of γ-H2AX/53BP1 foci, defined by their complexity, spatiotemporal behavior and repair characteristics. The data uncover that high-LET radiation beams generate highly complex γ-H2AX/53BP1 foci clusters with a larger overall size, increased irregularity and delayed resolution compared to low-LET γ-rays. Based on the finding that neon ions produce more complex γ-H2AX/53BP1 foci clusters than boron ions, 45% of which persist 24 h after irradiation, the authors hypothesize that the complexity of a DSB depends critically on the particle track-core diameter. Thus, different particles with similar LET may generate different types of DNA damage, which should be considered in future research [50].

Electron microscopy data obtained after exposure of human fibroblasts to carbon ions complement the data obtained with high-resolution confocal microscopy and ascertain the formation of highly complex and clustered DSBs. The dimensions of these clustered lesions along the particle tracks depend on chromatin density, as larger DSB clusters predominantly localize in condensed heterochromatin. High-LET IR produces clearly higher DSB yields than low-LET X-rays, with up to ~500 DSBs per μm^3^ track volume. A substantial fraction of these heterochromatic DSBs persist for longer periods of time suggesting difficulties in their repair. By contrast, DSBs induced by low-LET IR are efficiently rejoined within 24 h, both in eu- and heterochromatin. These data in aggregate support the hypothesis that the spacing and quantity of DSBs in clustered lesions affect DNA repair efficiency, and may determine the radiobiological outcomes [51].

A further study employed immunofluorescence analysis to follow the accumulation of endogenous 53BP1 and replication protein A (RPA) on chromatin after exposure to X-rays and α-particles [52]. Notably, 53BP1 foci induced by α-particles contain multiple RPA foci, suggesting multiple resection events, which are not observed following X-ray irradiation.

## 3. Exposure to High-LET IR Activates Signaling Networks Mainly Regulated by ATR

A prerequisite for the efficient processing of DSBs, including complex and clustered DSBs, is the activation of DDR. DDR involves the recognition of the lesion (DSBs in our case), followed by the initiation of cellular signaling cascades that activate, among others, cell cycle checkpoints and promotes DNA repair. In parallel, cells initiate other responses, such as the modulation of chromatin structure and transcription. These responses may be local and limited to the immediate vicinity of the DSB, or global affecting the entire cell [53], and may ultimately lead to apoptosis or senescence. Three apical protein kinases, members of the family of phosphoinositide 3-kinase (PI3K)-related kinases (PIKKs)—ATM, ATR and DNA-PKcs—are in the center of DDR activation. In the next paragraphs, we discuss the interplay between ATM and ATR in the regulation of G_2_ checkpoint and DNA end resection following exposure to low- and high-LET IR. The function of DNA-PKcs in the regulation of G_2_ checkpoint is not the subject of this review.

Ataxia telangiectasia mutated (ATM), the vertebrate orthologue of budding yeast protein kinase Tel1, is a primordial apical kinase that is commonly considered central to the orchestration of the cellular responses to DSBs [54,55]. ATM gets rapidly activated in response to DSBs and oxidative stress and is recruited to chromatin by interacting with the carboxy-terminus of NBS1 [56]. ATM undergoes autophosphorylation, but also phosphorylates a variety of downstream substrates and effector kinases, to coordinate DSB- repair, apoptosis, checkpoint activation and transcriptional arrest, as well as a great variety of additional cellular processes [53]. The pattern of ATM activation, as revealed by following its autophosphorylation at pS1981, shows a sharp peak in close proximity to DSBs. This contrasts the pattern of γ-H2AX, one of the ATM’s downstream substrates, which shows a relatively even distribution proximate and distal to the breaks and suggests that the active form of ATM is predominantly located near DSBs [57].

Ataxia telangiectasia-mutated and Rad3-related (ATR) is the human orthologue of the budding yeast protein Mec1. Unlike ATM and DNA-PKcs, it is essential in proliferating cells and is mobilized by a broad variety of genotoxic insults. Common characteristic of all is the generation of DNA single stranded regions (ssDNA). ATR is recruited by its interacting partner ATRIP to long stretches of ssDNA after they have been coated by the heterotrimeric ssDNA-binding protein complex RPA [58]. Important and relevant here is the ATR activation induced at resected DSBs.

A recent study from our group [59] showed that at low IR-doses that induce low DSB-numbers in the genome, ATM and ATR epistatically regulate the G_2_ checkpoint, with ATR at the output-node, interfacing with the cell cycle predominantly through CHK1 (Figure 3). Strikingly, at low IR-doses, ATM and ATR epistatically also regulate resection, and inhibition of either activity fully suppresses resection. At high IR-doses that induce high DSB-numbers in the genome, the tight ATM/ATR coupling relaxes and independent outputs to G_2_ checkpoint and resection occur. Consequently, both kinases must be inhibited to fully suppress checkpoint activation and resection.

Intriguingly, our more recent results illustrate that cells exposed to low doses of α-particles and iron ions show an enhanced G_2_ checkpoint response, which is mainly regulated by ATR. Conversely, ATM and ATR regulate cooperatively the G_2_ checkpoint after exposure to low doses of X-rays (Figure 3) [60]. These findings are in line with a previous report [61] showing that ATR plays a more important role following irradiation with carbon ions, as opposed to X-rays. Another study also provides evidence that complex and clustered DSBs generated upon exposure of tumor cell lines to carbon ions results in enhanced DNA end resection associated with enhanced activation of ATR. Chemical inhibition of ATR using the small molecule inhibitor, VE-821, causes abrogation of G_2_/M checkpoint and forces the transition of cells into mitosis [62]. These findings suggest an increased engagement of resection-dependent DSB repair pathways in the processing of complex and clustered DSBs generated after exposure to high-LET IR.

Efficient cellular responses to DSBs require the integration of multiple factors, coordinated by various post-translational chromatin modifications and chromatin-associated proteins [63]. Notably, the post-translational modifications of histones have been shown to play an essential role in initiating and regulating cellular responses to DSBs.

A study published in 2017 ascertains increased ubiquitylation of histone H2B at lysine 120 following exposure to high-LET radiation, but not to low-LET X-rays. This modification is very frequent in embryonic stem cells and is associated with transcriptional activation of multiple genes in the process of cellular differentiation [64]. This specific post-translational modification is catalyzed either by the RNF20/RNF40 complex, or by the multisubunit histone acetyltransferase complex subunit two (MSL2) and is important for the cell viability after exposure to high-LET IR. It could also affect histone H3 methylation at lysine 4 and could reduce H3 acetylation at lysines 9 and 56, which are post-translational modifications causing chromatin relaxation and possibly increased mobility of DNA ends to facilitate rejoining.

As indicated above, the prompt activation of DDR following exposure to IR frequently integrates distinct cellular mechanisms that are also responsible for the initiation of apoptosis. Given that many cancer types develop mechanisms to suppress apoptosis, it is of great importance to uncover alternative ways for apoptosis enhancement. Several reports indicate that high-LET IR induces enhanced levels of apoptosis, when compared to low-LET IR. Flow cytometry analysis of Annexin V- and 7-amino-actinomycin-positive cells exposed to different doses of γ-rays or boron ions reveals a three-fold increase in apoptosis 48 h after exposure to high-LET ions as compared to low-LET IR [50]. Another report came to similar conclusions [65] and demonstrated increased levels of late apoptosis in hematopoietic and progenitor cells exposed to low doses of neutrons. Neutron irradiation also increases the rates of early apoptosis in osteosarcoma cells, as compared to γ-rays and is associated with enhanced Caspase-3 and Caspase-9 expression and cleavage, PARP-1 cleavage and increased reactive oxygen species production [66]. Interestingly, there is increasing evidence that high-LET IR induces apoptosis in cancer cells regardless of p53 status [67,68]. When cells expressing mutated forms of p53 are irradiated with high-LET IR, Caspase-3 cleavage and activation is observed, followed by PARP1 cleavage. Notably, a Caspase-9 inhibitor suppresses Caspase-3 activation and apoptosis induction to a greater extent than Caspase-8 inhibitor after high-LET IR. These results suggest that high-LET IR enhances apoptosis by activation of Caspase-3 through Caspase-9, independently from p53 status, and might be relevant for apoptosis induction in p53-deficient tumors.

## 4. DSB Repair Pathways

A key target of DDR signaling is the regulation of DSB repair that is essential for the restoration of DNA sequence and structure to safeguard genome stability. Higher eukaryotes have evolved four major DSB repair pathways, which have distinct cell cycle and DNA sequence homology dependencies and operate with diverse fidelity and kinetics [69,70]. c-NHEJ is the predominant DSB repair pathway. It operates throughout the cell cycle without requirements for homology, although it occasionally utilizes microhomologies of 1–4 bp. Homologous recombination (HR) exhibits pronounced cell cycle dependence and operates only in S- and G_2_-phase of the cell cycle when a sister chromatid becomes available after DNA replication. HR requires long stretches of homology—typically more than 100 bp. Alternative end-joining (alt-EJ) typically requires microhomology of at least 2 bp (usually more) and less than 20 bp. Single-strand annealing (SSA) requires longer tracts of homology than alt-EJ that may occur on the same DNA molecule. A major discriminating characteristic between c-NHEJ and all remaining DSB repair pathways is its independence of resection at DSB ends. As a consequence, DSB repair pathways are frequently classified as resection dependent (HR, alt-EJ and SSA) and resection independent (c-NHEJ). In the following sections, we describe the key features of each of these DSB repair pathways, emphasize their inherent propensities for errors and describe the types and sources of errors they can produce.

### 4.1. Classical Non-Homologous End Joining

The major pathway for the repair of DSBs in both proliferating and quiescent cells is c-NHEJ. It is initiated by the binding of KU70/80 heterodimer to DSB ends, followed by the association of DNA-dependent protein kinase catalytic subunit, DNA-PKcs, to form the DNA-PK holoenzyme. DNA-PK is thought to serve as a landing pad for the recruitment of other c-NHEJ factors. The ligation of the broken DNA termini is catalyzed by the DNA ligase 4 (LIG4)-XRCC4 complex, an activity that is facilitated by XRCC4-like factor (XLF) and/or by paralogue of XRCC4 and XLF (PAXX) [71]. It is commonly believed that prior to ligation, DNA-PKcs must be displaced from DNA ends by autophosphorylation. Under certain circumstances, the generation of ligation-compatible ends requires DNA end processing, which can include excision, modification or addition of nucleotides. This is particularly important for IR-induced DNA ends that are typically non-ligatable (Figure 1). Activities important for DNA end processing include the Artemis nuclease, the polymerases-μ (Polμ) and λ (Polλ), the tyrosyl-DNA phosphodiesterase 1 (TDP1) and the polynucleotide kinase 3′-phosphatase (PNKP). The DNA end configuration at a DSB determines which of these factors are required for preparation of DNA ends for ligation.

Artemis and TDP1 can remove 3′-phosphoglycolates that are produced after IR and block ligation (Figure 1), whereas PNKP adds a 5′-phosphate to ends that have lost it after IR to generate ligatable ends [72,73]. It is likely that the presence of additional DNA lesions (e.g., base damage) at the DNA ends, in a CDS site, will affect the initial synaptic steps of c-NHEJ—for example by also recruiting components of BER to the site.

In contrast to HR, which usually restores the original sequence in the vicinity of the DSB using sequence information from the undamaged sister chromatid, c-NHEJ restores the integrity of the DNA, but not its original sequence. Indeed, c-NHEJ has no built-in mechanisms ensuring the restoration of DNA sequence at the DSB. However, the high processing speed is considered its most prominent and important feature, as it maximizes the probability of joining of the original DNA ends—by reducing the time available for diffusion of the two DNA ends away from each other. In summary, distinct forms of DNA ends require different types of end-processing before ligation, generating a variety of repair junctions that are frequently associated with small deletions or insertions. However, accurately joined products are also detected, especially when the DNA ends are compatible for direct ligation—e.g., staggered ends generated by restriction endonucleases [74,75].

### 4.2. Homologous Recombination (HR)

HR is the only error-free DSB repair process [76,77] and can be divided into three stages: pre-synaptic, synaptic and post-synaptic. It starts with the sensing of the DSB by the MRN (MRE11-RAD50-NBS1) complex, which, in cooperation with CtBP-interacting protein (CtIP), first generates a nick in one DNA strand near the 5′ end and then degrades the resulting fragment in a 3′ to 5′ direction, thereby creating a short 3′-overhang. The nucleases Bloom syndrome protein (BLM)–DNA replication ATP-dependent helicase/nuclease DNA2 and exonuclease 1 (EXO1) can mediate longer resection in a 5′ to 3′ direction to form an extended tract of ssDNA with 3′-overhangs. The 3′-tails are rapidly covered by RPA, which protects the resulting ssDNA from nucleolytic degradation and the formation of secondary structures. Through the mediation of BRCA2 in concert with PALB2 and the BRCA1–BARD1, RPA is displaced by RAD51 in the ssDNA to form a nucleoprotein filament. This process is facilitated by the RAD51 paralogs (RAD51B, RAD51C, RAD51D, XRCC2 and XRCC3) [78,79,80].

During synapsis, the RAD51 nucleoprotein filament searches for homology in the sister chromatid and invades the dsDNA to form a Holliday junction—a key intermediate in HR [81,82]. RAD54 promotes DNA synthesis and branch migration by dissociating RAD51 from heteroduplex DNA. In the post-synaptic steps, the extended Holliday junction is resolved in a variety of ways that define specific HR sub-pathways [83].

It is believed that the synthesis-dependent strand annealing (SDSA) is the most relevant subpathway in the repair of IR-induced DSBs by HR. In this final step, the newly synthesized strand anneals with the similarly processed second DNA strand to restore integrity in the molecule, and the process is completed by DNA synthesis and ligation [84,85].

HR can accommodate a wide spectrum of structural DNA end substrate configurations at the DSB, such as variations in the overhang length, DNA end sequence and DNA end chemistry. The unique templated nature (through the sister chromatid) of DSB repair by HR not only ensures the structural restoration of the DNA molecule, but also enables the preservation of DNA sequence at the DSB. As a result, HR is the only error-free repair pathway on all counts.

### 4.3. Alternative End-Joining (Alt-EJ)

The alt-EJ, also known as a backup non-homologous end-joining (b-NHEJ) [86,87,88], microhomology-mediated end joining (MMEJ) [89] or theta-mediated end joining, benefits from short (2 to 20 bp) stretches of microhomology that are exposed following limited processing of DSB ends. In the first step PARP1 recognizes the DSB [90,91] and promotes short-range DNA end resection by CtIP and the MRN complex—a step shared with HR. The repair process continues with the bridging and annealing of 2–20-bp (most often 3–8-bp) microhomologies in 3′ tails, facilitated by DNA POLθ and the unpaired non-homologous 3′ tails are digested by the ERCC1/XPF nuclease. Resulting gaps within DNA strands are next filled-in by POLθ-mediated DNA synthesis [92], and DSBs ends are rejoined by the DNA Ligase 3 (LIG3)/XRCC1 complex [93,94,95]. In the absence of the more efficient LIG3, DNA Ligase 1 (LIG1) can also take over to catalyze the final step of DNA ligation [96]. Alt-EJ operates with slower kinetics and lower efficiency than c-NHEJ and as a consequence is more error-prone. Thus, deletions and other modifications at the junction are larger than after c-NHEJ. It is also particularly relevant that during processing of DSBs with alt-EJ, the joining probability of unrelated ends is markedly increased. Thus, alt-EJ is considered a dominant source of structural chromosomal abnormalities (SCAs).

### 4.4. Single Strand Annealing (SSA)

Single strand annealing is a RAD51-independent mechanism that bridges two homologous 3’-ssDNA ends (for example, at tandem repeats), which results in obligate deletion of the interstitial fragment between the repeats [97]. Therefore, when repairing IR-induced DSBs, SSA is more error-prone than c-NHEJ or alt-EJ. SSA requires extensive DNA end resection and RPA displacement to reveal complementary homologous sequences [98,99]. SSA requires RAD52 for the annealing step, the structure-specific endonuclease XPF–ERCC1 for removal of unpaired non-complementary tails and LIG1 for ligation of the remaining nick [98,100].

## 5. Specific Processing Characteristics of High-LET-Induced DSBs and DSB Clusters

### 5.1. DSB Clusters Compromise c-NHEJ

It is well documented that the predominant pathway involved in the processing of low-LET IR-induced DSBs is c-NHEJ [69,71]. However, there is evidence that DSB clusters, generated after exposure to high-LET IR and in defined biological models, suppress c-NHEJ [49,101]. Although the effect of c-NHEJ suppression after high-LET IR is clearly observed for the subset of DSBs determining cell survival, it is expected to partly pertain for the overall processing of DSBs. Indeed, Western blot analysis shows lower expression levels of key c-NHEJ enzymes (KU70/KU80 and DNA-PKcs), 24 h after exposure to high-LET neutrons, as compared to non-irradiated controls or cells irradiated with γ-rays [56]. Moreover, pulse-field gel electrophoresis analysis, following DSB repair in c-NHEJ mutants exposed to iron ions fails to reveal contributions of c-NHEJ to the repair process [19]. However, there are reports, utilizing c-NHEJ and HR mutants, suggesting that the predominant role of c-NHEJ remains unaffected following exposure to high-LET IR [102]. These apparently contradictory results generate uncertainties, which necessitate in-depth studies on high-LET IR effects on DSB repair and a quantitative evaluation of the contribution of c-NHEJ, HR, alt-EJ and SSA to DSB processing. In this analysis, it will be useful to study in addition to total DSBs, also different subsets of DSBs: DSBs that break chromosomes, DSBs responsible for cell killing, etc.

### 5.2. Engagement of Resection-Dependent DSB Repair Pathways

Suppression of c-NHEJ owing to changes in the complexity of DSBs would bring to the fore resection-dependent DSB repair pathways, including HR. Indeed, several studies describe a LET-dependent shift from c-NHEJ, dominating at low-LET IR, to increased contribution of HR at high-LET IR [52,103,104,105]. The high-LET dependent stimulation of DNA end resection is confirmed by immunofluorescence and live-cell imaging analysis following the accumulation of endogenous 53BP1 and replication protein A (RPA) on chromatin after exposure to X-rays and α-particles [52]. The results reveal that, in contrast to α-particle-induced 53BP1 foci, X-ray-induced foci resolve quickly and more dynamically, as they show an increase in size 8 h post-irradiation. However, 53BP1 foci generated after exposure to α-particles resolve slower and less dynamically, as their size remains similar over time. However, the initial average focus area following exposure to α-particles is bigger, compared to the area after exposure to X-rays. In addition, the number of individual 53BP1 and RPA foci is higher after X-ray irradiation, whereas the total focus intensity is higher after α-particle irradiation.

As already discussed, 53BP1 foci induced by α-particles contain multiple RPA foci, suggesting multiple resection events, which are not observed following X-ray irradiation. Similar studies follow the recruitment of resection factors to DSBs generated after exposure to a broad spectrum of charged particles. The reported findings demonstrate an enhanced accumulation of RPA, ATR, MRE11, CtIP and EXO1, as compared to X-ray-generated DSBs [106].

Interestingly, extended cell cycle analysis demonstrates that resection-promoting factors are present at DSBs generated by high-LET IR, even in G_1_-phase where resection is normally suppressed. In line with this observation, another study reports that complex DSBs generated in G_1_-phase cells by α-particles are substrates of resection-dependent repair pathways utilizing the exonuclease activities of MRE11 and EXO1 [107,108]. This report also shows that under these conditions DNA end resection does not promote HR, but rather stimulates a specific form of resection-dependent c-NHEJ.

### 5.3. Contribution of Homologous Recombination to the Processing of DSB Clusters

DNA end resection-dependent stimulation of DSB repair pathways following exposure to high-LET IR suggests that complex and clustered DSBs can be repaired by HR [103]. This phenomenon is, indeed, observed in experiments with animal models employing wild-type and HR-deficient mice, as well as in different HR mutants generated in DT40 cells [49]. The respective studies demonstrate that HR mutants are more sensitive to high-LET IR than to low-LET IR. In contrast, c-NHEJ-deficient mice or DT40 cells are equally sensitive to both IR modalities. Moreover, the authors demonstrate that the distinct binding properties of MRE11 and KU underpin the different efficiencies of HR and c-NHEJ in the repair of high-LET-induced DSBs [49].

Other studies confirm the recruitment of RAD51 and factors involved in DNA strand invasion [109] to sites of clustered DSBs. Similar results have been reported for cells exposed to proton irradiation [110,111]. HR-deficient rodent cells with a deletion in RAD51D also demonstrate strong dependence on HR for survival after exposure to iron ions [103]. These conclusions are supported by results of similar experiments utilizing human cells, deficient in RAD51D or RAD51. Notably, cells exposed to iron ions in S-phase are radioresistant, which is in line with the observation that high-LET-induced DSBs are repaired by HR. Notably, mammalian cells deficient in RAD51 paralogs are also sensitive to HI [103].

### 5.4. Promotion of Error-Prone Alt-EJ by DSB Clusters Results in Enhanced Genomic Instability

Although the accumulated data suggest that HR is involved in the processing of high-LET IR-induced DSBs, it is unlikely that HR is the main contributor to the processing of high-LET induced DSBs, as HR is error-free while high-LET IR is associated with dramatic increases in SCAs that are generated by error-prone DSB repair pathways (see above). Indeed, we have previously demonstrated that the inhibition of c-NHEJ by high-LET IR results in a shift to PARP-1-dependent alt-EJ that is characterized by increased formation of SCAs [36]. Moreover, high-LET IR dramatically increases the incidence of SCAs as compared to X-rays in human lymphocytes [112] and their hematopoietic progenitors [113]. Extreme proximity of DSBs at a cluster also favors SCAs [51]. The employment of cytogenetic techniques, such as G_2_-premature chromosome condensation (G_2_-PCC) and multicolor fluorescence in situ hybridization (mFISH), also confirms that high-LET IR is more effective at inducing SCAs when compared to low-LET IR [114,115,116].

Cells harboring unrepaired breaks that are transmitted through the cell cycle due to a lack of proper G_2_-checkpoint activation, propagate chromosomal aberrations or undergo mitotic catastrophe [117]. Increased incidence of SCAs has also been observed following induction of enzymatic DSB clusters [36,118]. It is associated with increased engagement of alt-EJ. Recent investigations confirm the role of PARP-1 in the processing of complex DSBs inflicted after exposure to α-particles, as well as low-energy, higher-LET protons. Interestingly, this study also reveals a role for deubiquitylation in stabilizing PAPR-1 protein levels by preventing its degradation through the proteasome [119]. Since PARP-1 is a major factor implicated in alt-EJ, these findings confirm the enhanced contribution of this repair pathway to the processing of complex and clustered DSBs induced after exposure to high-LET IR.

In summary, the data suggest that DSB repair, relying on first line DSB-processing pathways (c-NHEJ and to some degree possibly HR) is compromised within complex DSBs and DSB clusters, presumably through the associated chromatin destabilization, leaving alt-EJ as the last option and SCAs induction as a natural consequence [118].

## 6. Concluding Remarks

IR modalities of different LET elicit distinct biological effects reflecting changes in DSB character and processing. However, what causes these changes? High-LET IR induces highly complex, as well as clustered DSBs that compromise the prevailing hierarchy in DSB repair programs and generate a shift from resection-independent c-NHEJ towards resection-dependent repair mechanisms. This is a clear adaptation cells make to necessities generated by the type of DSBs induced, and which dictate their processing [70]. It manifests with the engagement of lower fidelity repair pathways, presumably because of their lower overall operational requirements for chromatin stability, and their ability to deal with DNA ends that have drifted away from each other for several reasons, including the severity and extent of breakage.

A plausible model of DSB repair pathway engagement integrating known biological responses to low- and high-LET IR for cells irradiated in G_2_-phase (selected because all DSB repair pathways are active) is shown in Figure 4. According to this model, when cells are exposed to low doses of low-LET IR, c-NHEJ and HR contribute almost equally to DSB repair [120]. The shift at low doses to error-free HR allows restoration of the genome with maximum fidelity. Mutagenic DSB repair processes including c-NHEJ, alt-EJ and SSA are partly or completely suppressed and only operate when HR fails to engage or complete. With increasing IR-dose of low-LET IR, HR is suppressed by mechanisms that remain to be characterized, while c-NHEJ clearly gains ground and becomes first choice. DNA end resection also remains active showing signs of suppression only above 20 Gy. Persisting resection under conditions of suppressed HR leads to increased engagement of error-prone alt-EJ and SSA (Figure 4).

On the other hand, when complex DSBs or DSB clusters are generated by exposing cells to high-LET IR, several profound changes in repair pathway balance take place. Perhaps the most profound is the suppression of c-NHEJ (at least for the subset of DSBs that determine cell survival) (Figure 4) [19].

Although the concept of high and low doses loses significance after exposure of cells to high-LET IR, as a first approximation, we develop our model as follows: At low doses of high-LET IR, repair of DSBs by c-NHEJ is suppressed and this allows increased DNA end resection. Resection will naturally stimulate HR, as well as alt-EJ and SSA. Based on the high induction of SCAs [60] in cells exposed to high-LET IR, we assume that a substantial proportion of DSBs is processed by alt-EJ and SSA and that therefore HR is at least indirectly partly suppressed (Figure 4). It remains an open question whether the documented chromatin fragmentation evoked by high-LET IR, owing at least partly to the induction of DSB clusters, elicits local (or global) increases in DNA end mobility that compromise c-NHEJ and HR, unleashing alt-EJ and SSA, which frequently restore integrity in the genome, but frequently also cause SCAs.

The information summarized in this review shows that significant advances have been made in our understanding of the mechanisms governing the biological effects of high-LET IR. However, it is also evident that large gaps of knowledge exist in the field that must be filled by future research. Thus, it will be important:To elucidate the parameters that determine the engagement of resection-dependent DSB repair pathways following exposure to high-LET IR and establish the connection between resection and the engagement of HR.To establish the factors and activities that determine the switch from high-fidelity HR to error-prone repair mechanisms as a function of dose and LET.To investigate whether active or passive changes or disruptions in chromatin organization in the vicinity of the DSB regulate the response with increasing LET and to identify the molecular underpinnings of candidate molecular mechanisms.To establish whether distinct chromatin marks drive/force/stimulate the switch to resection with increasing LET.To explore whether pharmacological targeting or genetic modulation of such activities offers means to improve the clinical application of different IR modalities in cancer therapy and radiation protection during space travel.

The field is mature and ready to explore new horizons in the mechanistic understanding of radiation response, while heavily drawing from the explosion of knowledge on the biological mechanisms of DDR and recent progress in the application of particle Physics in Radiation Medicine. At a time when space exploration quickly develops to a necessity and a routine and radiation therapy is actively seeking to maximize its potential, support in this endeavor appears more likely than ever. The future is bright!

## Figures and Tables

**Figure 1 molecules-27-01540-f001:**
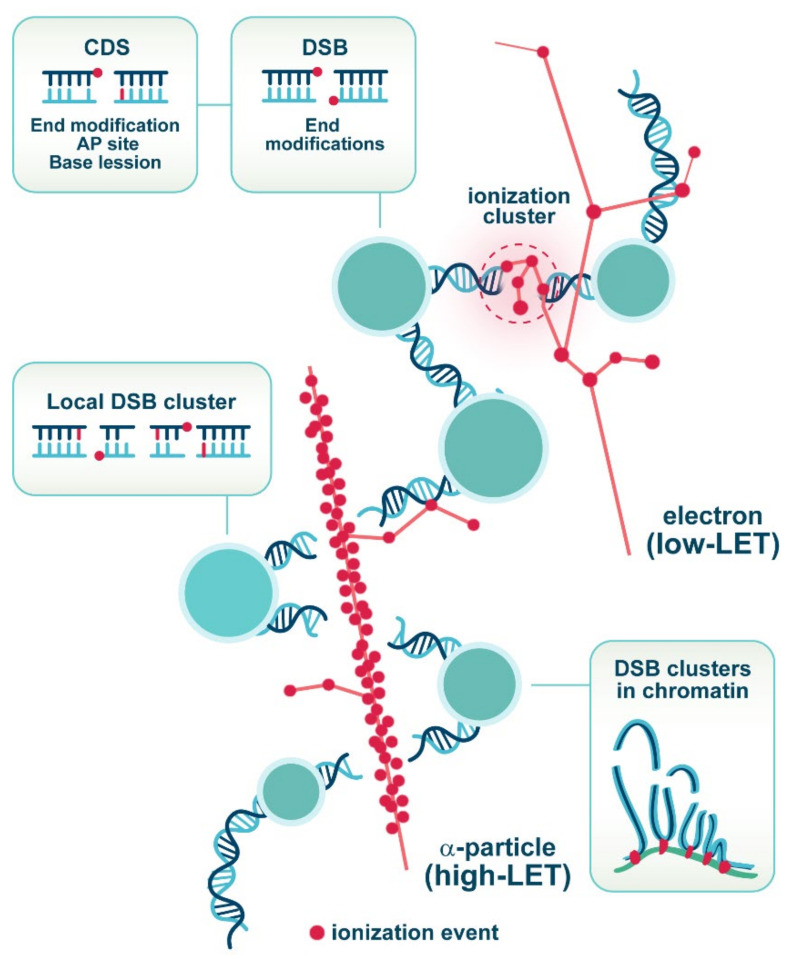
Generation of DSBs by low-LET electrons and high-LET α-particles. Ionization events along the radiation tracks are presented as red dots, while the DNA molecule is rendered as part of chromatin organized with nucleosomes. The formation of DSBs of different complexity is induced by both low- and high-LET IR and these DSBs frequently harbor chemically modified DNA ends that are not directly ligatable and require additional processing. Clustered damage sites (CDS) of different damage permutations are detectable after low-LET IR and their number increases with increasing LET. Notably, high-LET IR generates with significantly higher probability DSB clusters, comprising multiple DSBs located in close proximity along the DNA that destabilize chromatin and, thus, the processing of the individual DSBs that always takes place in the context of chromatin.

**Figure 2 molecules-27-01540-f002:**
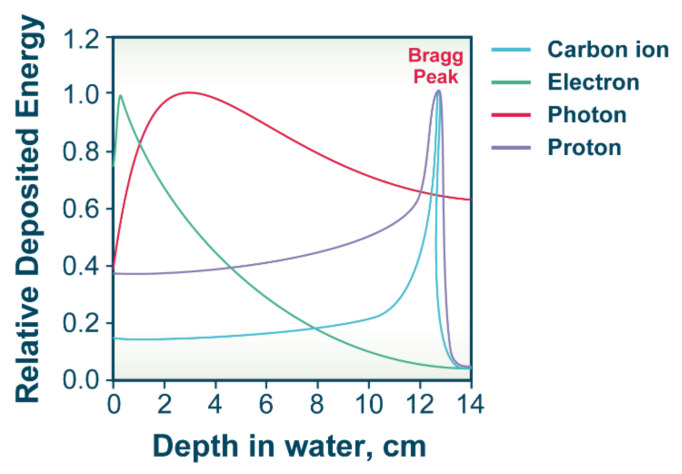
Energy deposition of different IR modalities. Idealized plots representing the energy deposition pattern of carbon ions, electrons, photons and protons in water. The main characteristic of charged particles is that they deposit their energy following a pattern known as the “Bragg peak”.

**Figure 3 molecules-27-01540-f003:**
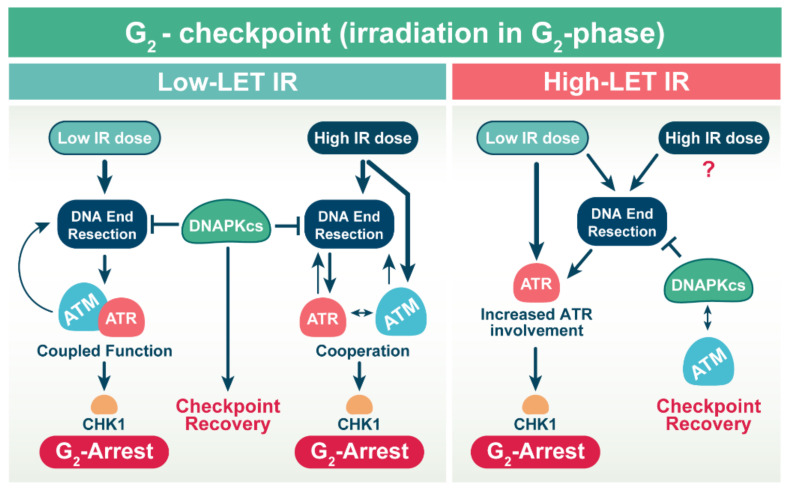
Activation of DNA damage checkpoint in cells irradiated in G_2_-phase of the cell cycle. Exposure of cells to high-LET IR suppresses the repair of DSBs by c-NHEJ, which is associated with increased DNA end resection. Elevated DNA end resection activates ATR that now mainly regulates the G_2_ checkpoint (see text for details).

**Figure 4 molecules-27-01540-f004:**
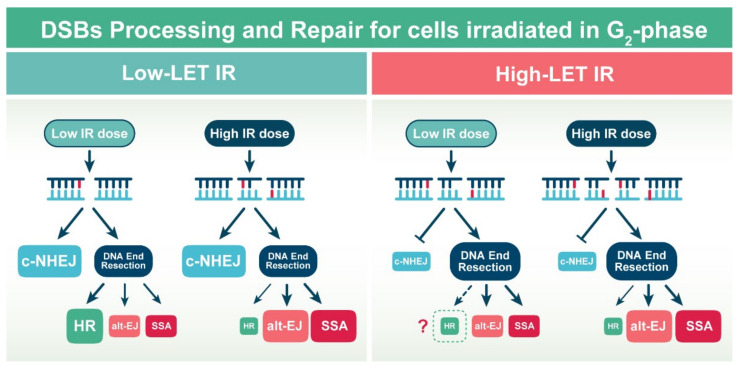
Exposure of cells to high-LET IR disrupts the balance between DSB repair pathways by shifting the processing to DNA end resection-dependent mechanisms (see text for details).

## Data Availability

Not applicable.

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
