# Peer review of "DNA Damage Clustering after Ionizing Radiation and Consequences in the Processing of Chromatin Breaks"

_molecules, 2022, doi:10.3390/molecules27051540_

Round 1
Reviewer 1 Report
Comment for authors:
In this review article, authors have summarized the recent knowledge about low / high-LET ionizing radiation (IR) induced DNA damage complexity and consequential DNA damage response, repair and cellular homeostasis. The authors have given special emphasis on the specific cellular DNA damage responses (DDR) elicited by high-LET IR and compare them to those of low-LET IR and followed by description about how distinct form of DSBs modulate the interplay of DSB-repair pathways. Overall, authors have nicely compile existing information low / high-LET ionizing radiation (IR) induced DNA damage response. Additionally, authors have published a series of research articles justifying review theme.
Thus, the review should be considered for publication with some minor changes.
Minor comments:
1. Line no. 93-94, authors should provide the patents update till 2021.
2. Line no. 98-99, provides source of information.
3. Line no. 100-101, simplify the title language.
4. Line no. 104, Authors may cite more appropriate reference.
5. Line no. 111-114, citation of reference is needed.
6. Line no. 188, describe term “chromothripsis”.
7. Line no. 201, describe term “RBE”.
8. Cite appropriate reference for the line 266-269 and 269-273.
9. Cite appropriate reference for the line 410-413.
Author Response
Response to Reviewers comments (molecules-1600583)
Response to Reviewer 1
“Comment for authors:
“In this review article, authors have summarized the recent knowledge about low / high-LET ionizing radiation (IR) induced DNA damage complexity and consequential DNA damage response, repair and cellular homeostasis. The authors have given special emphasis on the specific cellular DNA damage responses (DDR) elicited by high-LET IR and compare them to those of low-LET IR and followed by description about how distinct form of DSBs modulate the interplay of DSB-repair pathways. Overall, authors have nicely compiled existing information low / high-LET ionizing radiation (IR) induced DNA damage response. Additionally, authors have published a series of research articles justifying review theme. Thus, the review should be considered for publication with some minor changes.”
We thank the reviewer for the positive response and for considering our manuscript for publication in Molecules, Special Issue “The Beauty of Clustered DNA Lesion”. We appreciate the time and effort the reviewer has dedicated to provide feedback on our review article and to suggest improvements, which increase the readability and clarity of the article.
“Minor comments:
- Line no. 93-94, authors should provide the patients update till 2021.
2. Line no. 98-99, provides source of information.
3. Line no. 100-101, simplify the title language.
4. Line no. 104, Authors may cite more appropriate reference.
5. Line no. 111-114, citation of reference is needed.
6. Line no. 188, describe term “chromothripsis”.
7. Line no. 201, describe term “RBE”.
8. Cite appropriate reference for the line 266-269 and 269-273.
9. Cite appropriate reference for the line 410-413.”
In the revised version of the manuscript we have addressed all minor comments indicated by the reviewer. We have updated the information about the patients treated with charged-particle radiotherapy up to 2020 (there is no available information for 2021) and have incorporated additional references at the paragraphs, suggested by the reviewer. We have simplified the language expressions, when appropriate, and have updated the definitions of “chromotripsis” and “RBE”.
We hope that the corrections improved the manuscript and have made it more comprehensive to the reader.
Reviewer 2 Report
The manuscript by Mladenova et al. provides a comprehensive review differential responses to clustered DNA double strand breaks (DSBs), typically generated by high-LET ionising radiation, as compared to “conventional” DSBs. It combines an extremely useful update on this important topic with insightful analysis of the underlying molecular and cellular mechanisms and proposes refined models to integrate the state-of-art knowledge. The review thus contains a significant conceptual advance, which will make it a useful reading for the experts in the field. At the same time, thanks to the thorough and clear introduction, this article will be understandable to readers from different research fields, physicians and students. The topic fits perfectly to the special issue “The Beauty of Clustered DNA Lesion”.
In all, the authors have done a very good work and I recommend to accept this manuscript for publication in Molecules. I only have minor suggestions, which the authors may wish to follow.
Specific points:
- Perhaps it would be more appropriate to abbreviate charged-particle radiotherapy as “CPRT” (for the sake of uniformity with other discussed radiotherapy technologies (IGRT, IMRT, SBRT, and so on).
- References are missing in the paragraph on lines 54-58; to the sentence on line 99; to the fragment on lines 269-273. Also, it would be extremely helpful to recapitulate the most important references in the section 6. “Concluding remarks”.
- Could “DSB-clusters” (lines 183-194 and the legend to Figure 1) be defined with more precision, it particular for the sake of a clear discrimination between “clustered damage” and “clustered DSB”?
- It does not seem that the statement “available results suggest that high-LET IR 215 compromises the efficiency of classical non-homologous end-joining (c-NHEJ)” follows from the paragraph content (lines 211-218).
- “Substantial fraction of these heterochromatic DSBs persist for longer periods of time suggesting difficulties in their repair.” (lines 238-240). Would it be possible to specify more precisely the half-life of DSBs in heterochtomatin in hours or days?
- Perhaps it would be helpful to divide section 5 of the manuscript into several subsections (e,g, difficulties of c-NHEJ – discussion of HR – discussion of alt-EJ)
- The last paragraph ends with the necessity to close the existing gaps of knowledge in the field. It would be extremely helpful to specify these gaps here and outline the priority tasks, where the future research effort should be directed.
Suggested text corrections:
line 12: delete “as compared to low-LET photons” (“unique” does not need a comparison)
line 23: “consequences on” should better read “consequences for” or “impact on”
line 111: please check whether “linear quadratic” is correct
line 201: delete “measured”
line 326-327: consider joining the paragraphs
line 471-472: is there a difference between “quickly” and “more dynamically” in the context?
line 508-509: connect paragraphs
The following sentences and text fragments may need re-phrasing for better understanding:
Lines 42-43
Lines 45-46
Lines 63-64
Lines 70-73
Lines 275-277
Lines 294-297
Lines 386-388
Lines 536-537
Author Response
Response to Reviewers comments (molecules-1600583)
Response to Review 2
“The manuscript by Mladenova et al. provides a comprehensive review differential response to clustered DNA double strand breaks (DSBs), typically generated by high-LET ionising radiation, as compared to “conventional” DSBs. It combines an extremely useful update on this important topic with insightful analysis of the underlying molecular and cellular mechanisms and proposes refined models to integrate the state-of-art knowledge. The review thus contains a significant conceptual advance, which will make it a useful reading for the experts in the field. At the same time, thanks to the thorough and clear introduction, this article will be understandable to readers from different research fields, physicians and students. The topic fits perfectly to the special issue “The Beauty of Clustered DNA Lesion”.
In all, the authors have done a very good work and I recommend to accept this manuscript for publication in Molecules. I only have minor suggestions, which the authors may wish to follow.”
We thank the reviewer for the strongly positive evaluation of our work and for the time and effort invested to provide feedback, insightful comments and valuable suggestions.
“Specific points:
- Perhaps it would be more appropriate to abbreviate charged-particle radiotherapy as “CPRT” (for the sake of uniformity with other discussed radiotherapy technologies (IGRT, IMRT, SBRT, and so on).”
We have changed the abbreviation, as suggested by the reviewer. Additionally, to facilitate the reading, we have removed all abbreviations that were not repeatedly used in the text.
“• References are missing in the paragraph on lines 54-58; to the sentence on line 99; to the fragment on lines 269-273. Also, it would be extremely helpful to recapitulate the most important references in the section 6. “Concluding remarks”.”
All the missing references are now included in the revised version of the manuscript. This was a very useful suggestion!
“• Could “DSB-clusters” (lines 183-194 and the legend to Figure 1) be defined with more precision, it particular for the sake of a clear discrimination between “clustered damage” and “clustered DSB”?”
We thank the reviewer for raising this point. We now define the term precisely.
“• It does not seem that the statement “available results suggest that high-LET IR 215 compromises the efficiency of classical non-homologous end-joining (c-NHEJ)” follows from the paragraph content (lines 211-218).”
We clarify the passage.
“• “Substantial fraction of these heterochromatic DSBs persist for longer periods of time suggesting difficulties in their repair.” (lines 238-240). Would it be possible to specify more precisely the half-life of DSBs in heterochtomatin in hours or days?”
As indicated in the referenced article, DSBs after low-LET IR are repaired within 24 h and repair kinetics is slower for heterochromatic DSBs. Quantitative data are not included in the publication (particularly with respect to cellular responses to high-LET IR) and we clarify the point.
“• Perhaps it would be helpful to divide section 5 of the manuscript into several subsections (e,g, difficulties of c-NHEJ – discussion of HR – discussion of alt-EJ)”
We are very grateful for this excellent suggestion. In the revised version of the review, Section 5 is divided into four subsections. It is true that this structure substantially improves the section.
“• The last paragraph ends with the necessity to close the existing gaps of knowledge in the field. It would be extremely helpful to specify these gaps here and outline the priority tasks, where the future research effort should be directed.”
We thank the reviewer for this suggestion. In the revised manuscript, we specify tasks and suggest priorities to close gaps.
“Suggested text corrections:
line 12: delete “as compared to low-LET photons” (“unique” does not need a comparison)
line 23: “consequences on” should better read “consequences for” or “impact on”
line 111: please check whether “linear quadratic” is correct
line 201: delete “measured”
line 326-327: consider joining the paragraphs
line 471-472: is there a difference between “quickly” and “more dynamically” in the context?
line 508-509: connect paragraphs
The following sentences and text fragments may need re-phrasing for better understanding:
Lines 42-43
Lines 45-46
Lines 63-64
Lines 70-73
Lines 275-277
Lines 294-297
Lines 386-388
Lines 536-537”
The revised manuscript incorporates all above suggestions. We have simplified the language, included missing references and improved the structure of some paragraphs.
- „line 471-472: is there a difference between “quickly” and “more dynamically” in the context?“
We thank the reviewer for pointing this out. In the cited article, „quickly” reflects the speed of 53BP1 foci resolution, whereas “dynamically” reflects the increase of foci size (but not number) over time – this is valid for foci formation and decay following exposure to X-rays; however, 53BP1 foci generated after exposure to a-particles resolve slower and “less” dynamically (according to the authors), because their size doesn’t change over time. However, the initial average focus area after exposure to a-particles is bigger. We explain this point in the corresponding passage.